# Experimental and Numerical Study of a Trapezoidal Rib and Fan Groove Microchannel Heat Sink

**DOI:** 10.3390/mi15060713

**Published:** 2024-05-28

**Authors:** Lufan Jin, Junchao Wang, Yixun Cai, Guangzhao Yang, Xuebing Hua, Zhenggeng Zhong, Xiao Pan, Chengyu Cai, Jia Qin, Mingxuan Cao

**Affiliations:** 1College of Optoelectronic Manufacturing, Zhejiang Industry and Trade Vocational College, Wenzhou 325003, China; 13819729461@163.com (X.H.); 15990722700@163.com (Z.Z.); caichengyu@zjitc.edu.cn (C.C.); qinjia@zjitc.edu.cn (J.Q.); 2Institute of Laser and Opto-Electronics, School of Precision Instruments and Opto-Eletronics Engineering, Tianjin University, Tianjin 300072, China; 3Penta Laser (Zhejiang) Co., Ltd., Wenzhou 325000, China; 4School of Mechanical and Automation Engineering, Wuyi University, Jiangmen 529020, China; w13137051728@163.com (J.W.); 13575282936@163.com (Y.C.); yangguangzhao1121@163.com (G.Y.)

**Keywords:** trapezoidal ribs, fan grooves, numerical simulation, experiments, thermal enhancement factor

## Abstract

A novel microchannel heat sink (TFMCHS) with trapezoidal ribs and fan grooves was proposed, and the microchannel was manufactured using selective laser melting technology. Firstly, the temperature and pressure drop at different power levels were measured through experiments and then combined with numerical simulation to explore the complex flow characteristics within TFMCHSs and evaluate the comprehensive performance of microchannel heat sinks based on the thermal enhancement coefficient. The results show that, compared with rectangular microchannel heat sinks (RMCHSs), the average and maximum temperatures of TFMCHSs are significantly reduced, and the temperature distribution is more uniform. This is mainly caused by the periodic interruption and redevelopment of the velocity boundary layer and thermal boundary layer caused by ribs and grooves. And as the heating power increases, the TFMCHS has better heat dissipation performance. When P=33 W and the inlet flow rate is 32.5 mL/min, the thermal enhancement factor reaches 1.26.

## 1. Introduction

Due to the rapid development of electronic devices towards integration and miniaturization, the heat flux of these devices has significantly increased. Highly integrated circuits and microelectronic devices are widely used in aerospace, intelligent manufacturing, and high-tech industries. These include applications such as cooling high-power semiconductor lasers, thermal management of aviation communication equipment, and heat dissipation of microelectronic devices. If the temperature increase in the device cannot be effectively managed, it may cause the device to malfunction, significantly impacting its stability and service life. Therefore, there is an urgent need for efficient heat transfer devices to eliminate the large amount of heat generated by electronic devices. One solution for effective cooling is the single-phase liquid-cooled microchannel radiator first proposed by Tuckman and Pisse in 1981 [1]. Compared with traditional radiators, it has higher heat transfer performance, smaller geometric dimensions, and lower coolant requirements.

At this point, a lot of papers have looked at the basic flow and heat transfer properties of single-phase convection in microchannel radiators. Among them, methods to enhance the heat dissipation performance of microchannels include nanofluids [2,3], inlet cross-section design [4,5], ribs and groove structures [6,7], porous media [8,9], wavy microchannels [10,11], etc. Adding ribs and groove structures to microchannels through flow destruction technology is an effective approach. K. Derakhshanpur et al. conducted numerical research on microchannels with semicircular ribs and found that an increase in the curvature of the semicircular ribs resulted in a significant increase in heat transfer coefficient and a slight increase in pressure drop [12]. Ergin Bayrak et al. used numerical simulation methods to study the heat dissipation of microchannels with different geometric shapes, discovering that microchannels with holes can mix the fluid at the center of the channel with the fluid near the wall, thereby improving heat transfer performance [13]. Datta et al. conducted a study on composite channels with ribbed grooves on the side walls. By comparing the Nusselt number, friction coefficient, and thermal performance changes, it was found that the composite channel had the best comprehensive performance [14]. Peitao et al. investigated the effects of rib height and cavity depth on the heat transfer performance of triangular ribbed microchannels. The optimal size of microchannels reduces the temperature difference at the bottom of the channel from 26 K to 17 K, resulting in an overall performance improvement of 1.2305 times [15]. 

However, while these numerical modeling efforts indicate that complex microchannel designs are constantly improving, fabrication of such heat sinks via conventional subtractive techniques (e.g., micromachining [16], laser processing [17], and anisotropic chemical etching [18]) has limitations. Advances in additive manufacturing (AM) technologies have made it possible to manufacture complex microchannels. Kirsch and Thole experimentally tested the wavy microchannels of additive manufacturing. The results indicate that the wall roughness generated by the AM process enhances heat transfer and also increases pressure loss [19,20]. Ganesan Narendran et al. used AM to manufacture single-layer and double-layer microchannel heat sinks. The influence of different materials on the performance of microchannels was explored and compared with traditional processing methods. The results showed that the 3D printing channel had a higher pressure drop and a larger Nusselt number [21]. Guang Pi et al. used selective laser melting (SLM) technology to manufacture microchannel structures with pits and cavities. Compared with straight microchannels, it increased the heat transfer performance by more than 10% [22]. Han et al. made a new type of biomimetic microchannel heat sink through 3D printing. The temperature difference inside the 3D-printed heat sink was 57% lower than that of the spider web heat sink, resulting in better temperature uniformity [23]. Joshi et al. conducted an experimental comparison between microchannel heat sinks manufactured with AM and those with conventional processing. The heat dissipation performance of AM microchannels improved by 46%, but the pressure drop increased by 91%, mainly due to the high surface roughness of AM microchannels [24].

As previously mentioned, numerous studies have demonstrated that the addition of rib and groove structures within microchannels through flow disruption techniques is an effective method to enhance microchannel performance. Selective laser melting is an effective technique for fabricating complex microchannels. In this study, we proposed a microchannel with an asymmetric rib and groove structure and fabricated the microchannel heat sink using SLM and CuCrZr powder. We conducted experiments to test the heat dissipation in the microchannels using deionized water as the working fluid. The thermal performance of the TFMCHS was compared with that of conventional straight channels to explore the feasibility of enhancing heat transfer. The results show that compared with the RMCHS, the average and maximum temperatures of the TFMCHS are significantly reduced, and the temperature distribution is more uniform. This is mainly caused by the periodic interruption and redevelopment of the velocity boundary layer and thermal boundary layer caused by ribs and grooves. And as the heating power increases, the TFMCHS has better heat dissipation performance. When P=33 W and the inlet flow rate is 32.5 mL/min, the thermal enhancement factor reaches 1.26.

## 2. Data reduction

We conducted experiments to examine the flow and heat transfer features at different flow rates, from 15 to 40 mL/min. The average Nusselt number (Nu) and heat transfer coefficient (h) are given by the following method:(1)$$Nu=\frac{hD_{h}}{\lambda_{f}},$$
(2)$$h=\frac{q_{w}A_{film}}{\Delta TA_{ch}L_{ch}}\cdot \frac{D_{h}}{\lambda_{f}},$$
(3)$$\Delta T=T_{w}-0.5(T_{in}+T_{out}),$$
where h, $q_{w}$, $A_{film}$, ΔT, and $A_{ch}$ are the heat transfer coefficient $W/\left(m^{2}\cdot k\right)$, the heating wall heat flux $W/m^{2}$, the bottom substrate heating area $m^{2}$, the temperature difference between the channel walls and fluid (K), and the contact surface area between the fluid and solid wall ${(m}^{2})$. $T_{w}$, $T_{in}$, and $T_{out}$ are the average temperatures of the heating film area and the inlet and outlet fluid temperature, respectively, all measured in K. 

The experimentally measured pressure drop across the microchannel is used to calculate the experimental friction factor given in Equation (4):(4)$$f_{ave}=\frac{\Delta PD_{h}}{2\rho_{f}Lu_{m}^{2}}.$$

The theoretical friction factor is predicted using the Hagenpoiseuille equation given by Equation (5):(5)$$f_{ave}=\frac{64}{Re}.$$

At the channel inlet, the following formula defines the Reynolds number (Re):(6)$$Re=\frac{\rho_{f}u_{in}D_{h}}{\mu_{f}},$$
where $u_{in}$ is the inlet velocity of the channel.

The following equations can be used to calculate the hydraulic diameter:(7)$$D_{h}=\frac{4W_{ch}H_{ch}}{2(W_{ch}+H_{ch})}.$$

The total thermal resistance of the heat sink and the pumping power is calculated as follows:(8)$$R_{th}=\frac{T_{\max}-T_{in}}{Q}=\frac{T_{\max}-T_{in}}{A_{film}\cdot q},$$
where $R_{th}$ is the thermal resistance of microchannel heat sinks, $T_{\max}$ is the maximum value of the temperature field, and $T_{in}$ is the inlet temperature.

The pumping power is expressed as follows:(9)$$W_{pp}=N\cdot A_{C}\cdot u_{in}\cdot \Delta p,$$
where $W_{pp}$ is the pumping power by the MCHS, N is the number of channels, $A_{C}$ is the cross-section area of each channel, and Δp is the total pressure drop across the microchannel.

The thermal enhancement factor (PEC) is defined as the ratio of the heat transfer coefficient of the enhanced channel to that of the non-enhanced channel at an equal pumping power and is given by [25]:(10)$$PEC={\left(\frac{Nu}{Nu_{0}}\right)/\left(\frac{f}{f_{0}}\right)}^{\frac{1}{3}},$$
where 0 represents the rectangular channel, i.e., a Rec.

## 3. Experiment and Simulation

### 3.1. Heat Sink Design

Figure 1 shows the isometric view of the straight-channel heat sink test piece and its dimensions. The channel height $H_{ch}$ is 1.1 mm, the channel width $W_{ch}$ is 1.1 mm, the channel length $L_{ch}$ is 20 mm, the wall thickness between the channels $W_{S}$ is 1.1 mm, the total width of the heat sink $W_{HS}$ is 10 mm, the total length of the heat sink $L_{HS}$ is 30 mm, and the total height of the heat sink $H_{HS}$ is 5 mm. 

The TFMCHS was created based on the conventional RMCHS. The design parameters of the TFMCHS are presented in Figure 2. The geometric structure of ribs and grooves is mainly designed with the goal of reducing the boundary layer. Among them, for the rib structure, the upper and lower edges of the right-angle trapezoid are 0.11 mm and 0.15 mm, respectively, and the height of the trapezoid is 0.15 mm. The depth of the fan-shaped groove is 0.2 mm, and the included angle is 30°. The spacing between the ribs and grooves is 0.8 mm, with a period of 2 mm. The rest of the dimensions of the TFMCHS are the same as those of the RMCHS. Figure 3 shows the isometric view of the TFMCHS test piece.

### 3.2. Processing of Microchannel Heat Sinks

This study used CuCrZr powder to prepare the microchannels. The particles are spherical and have a diameter between 30 and 50 µm. The microchannel heat sink was prepared using SLM technology, as shown in Figure 4a. Figure 4b shows the actual molding effect inside the microchannel, with $R_{z}$ values measured at the bottom surface of the microchannel ranging from 75 to 90 μm.

### 3.3. Preprocessing and Packaging of Microchannel Heat Sinks

Firstly, the conductive wire of the heating plate was connected to the DC power supply, and then the heating plate was fixed to the top surface of the microchannel heat sink using thermally conductive silicone. The thermal conductivity of the thermally conductive silicone was 5 $W/\left(m^{2}\cdot k\right)$. In order to achieve a good encapsulation effect for microchannel heat sinks, a dedicated positioning fixture was designed. The fixture was divided into upper and lower parts, which were fastened with two M4 screws to ensure a stable and water-tight testing process. A multichannel thermometer was used to measure the temperature curve of the bottom surface and the inlet and outlet of the radiator. Finally, the water pipe was tightly connected to the inlet and outlet of the heat sink. The fully encapsulated heat sink is shown in Figure 4c.

### 3.4. Experimental Procedures

A schematic diagram of the experimental apparatus is shown in Figure 5. It consisted of a test section, a power supply system, a data acquisition system, and a drainage system. Deionized water with a working fluid of 298.15 K was used. The ceramic heating plate was energized by a DC power supply with a range of 0–30 V and 0–10 A. After being pumped by a peristaltic pump, the coolant in the constant-temperature water tank entered the experimental section and carried away the heat of the heating plate through a microchannel heat sink. The heated coolant entered the water tank. The inlet water temperature was controlled at 298.15 k through a constant-temperature bath. The peristaltic pump had a filter at the front end to prevent pollutants from entering. The flow rate was accurately controlled within the range of 10~100 mL/min by adjusting the rotational speed of the peristaltic pump, with an uncertainty of 1.5%. Using a multichannel thermometer to measure the temperature at the inlet, outlet, and bottom of the microchannel, the uncertainty was 0.2 K. Using a pressure controller and pressure sensor to measure the pressure in the inlet and outlet sections, the uncertainty was 0.5%. These devices were connected to a PC for monitoring and recording temperatures and pressures during the experiments. 

During the experiment, packaging testing was carried out first. Then, the inlet flow rate was set through a peristaltic pump, the inlet water temperature was set through a constant-temperature water tank, and the required heating plate power was also set. After a stable water supply for half an hour, the experimental data were recorded.

### 3.5. Uncertainty Analysis

Due to errors in all the experimental data collected, a detailed uncertainty analysis was carried out on relevant parameters using the procedure described by Holman [26]. The uncertainty of each parameter was based on the uncertainty of the measured parameter, and these errors are listed in Table 1, respectively. According to the analysis, the uncertainty of the Nusselt number and friction coefficient were ±3.22% and ±3.94%, respectively.

### 3.6. Numerical Simulation

A three-dimensional numerical model of the microchannel heat sink was established using ANSYS FLUENT 16.0 software, and the convective heat transfer and fluid flow processes inside the microchannel were analyzed. The upper surface of the channel was in contact with the heating plate, and the remaining surfaces were set for natural heat dissipation. The pressure velocity was coupled using the SIMPLEC algorithm, and the momentum and energy equations were discretized using a second-order upwind scheme [27]. The entire microchannel was selected as the computational domain, and in order to simplify the problem, the following assumptions were made for the model: (1)The fluid is a Newtonian fluid, and the flow is laminar, incompressible, and steady-state [28];(2)Except for the viscosity of water, the properties of fluids and solids are constant [29];(3)Radiative heat transfer is neglected [30];
(11)∇⋅V=0
(12)$$\rho \left(v\cdot \nabla v\right)=-\nabla p+\nabla (\mu_{f}\cdot \nabla T_{f})$$
(13)$$\rho C_{P}v\cdot \nabla T_{f}=\nabla (\lambda_{f}\cdot \nabla T_{f})$$

For the solid region, the energy equation is
(14)$$\nabla (\lambda_{s}\cdot \nabla T_{s})=0.$$

Among them, V is the velocity of the fluid, p is the pressure in the fluid region, $T_{f}$ is the fluid temperature, and $T_{s}$ is the solid temperature. 

This simulation uses deionized water as the fluid and CuCrZr as the solid material. The material properties of water include a density of 1000 $\mathrm{kg}\cdot m^{-3}$, a thermal conductivity of 0.6 $w\cdot m^{-1}\cdot k^{-1}$, a constant pressure heat capacity of 4178 $j\cdot {\mathrm{kg}}^{-1}\cdot k^{-1}$, and a viscosity that varies with temperature. The material properties of CuCrZr include a density of 8960 $\mathrm{kg}\cdot m^{-3}$, a thermal conductivity of 288 $w\cdot m^{-1}\cdot k^{-1}$, and a constant pressure heat capacity of 377 $j\cdot {\mathrm{kg}}^{-1}\cdot k^{-1}$. 

The microchannel Inlet was set as a constant velocity inlet, the outlet was a pressure outlet, and the gauge pressure was zero. A uniform heat flux was applied at the top of the microchannel. Coupling boundary conditions were set on the fluid-solid coupling wall. 

In this study, the variation range of water temperature is 298.15–363.15 K, and the viscosity variation range is $3.17\times 10^{-4}-8.95\times 10^{-4}\mathrm{kg}/\left(m\cdot s^{-2}\right)$. The influence of visible temperature on viscosity cannot be ignored. Kestin et al. confirmed the relationship between temperature and viscosity in Equation (15) [31]:(15)$$log\left\{\frac{\mu (t)}{\mu \left(20\;{}^{\circ}C\right)}\right\}=\frac{20-t}{t+90}\left\{1.2378-1.303\times 10^{-3}(20-t)+3.06\times 10^{-6}(20-t{)}^{2}+2.55\times 10^{-8}(20-t{)}^{3}\right\}$$

The governing equations are regarded as converged once all residuals fell below $10^{-4}$. The heat sinks were meshed with hexahedron mesh. Grid independence studies were performed to ensure the numerical solution was not influenced by the number of grids. Four different numbers of grids were used, and the relative error between grid 4 and the average bottom temperature of other grids is shown in Table 2. The relative error between Grid 3 and Grid 4 was just 0.9%, so Grid 3 was used to obtain the numerical solution. 

## 4. Result and Discussion

### 4.1. Printing Equipment and Printing Parameters

Commercial laser-selective melting equipment (Dimetal-300, Leijia Additive Manufacturing Company, Guangdong, China) was used to print the designed microchannel heat sink samples. The printing parameters were set as follows: laser power at 350 W, scanning speed at 420 mm/s, scanning spacing of 0.06 mm, and layer thickness of 30 μm. To enhance printing precision and ensure structural integrity inside the microchannel heat sink, a 67° rotated printing method was chosen. After printing, a wire-cutting machine was used to remove the external support structures of the microchannel heat sink, as shown in Figure 4a. It can be observed that the external shape of the printed microchannel heat sink sample was well formed.

### 4.2. Verification of Simulation and Validation of Experiment for MCHSs

In order to verify the accuracy of the numerical solution, experimental data and simulation data of the microchannel were compared. Firstly, a multichannel thermometer was used to set five uniform temperature measurement points on the bottom of the microchannel to measure the average temperature of the bottom surface. The comparison between the experimentally measured temperature and the simulated temperature for MCHSs and TFMCHSs at $Q_{V}=30\;\mathrm{mL}/\min$ is shown in Figure 6. It can be seen that under different heating plate powers, the experimental results and simulation results are in good agreement, and the deviation of the average temperature of the radiator bottom surface is less than ±3.7%. The average temperature calculated by simulation is greater than the average temperature tested by experiments. This is because the simulation calculation only considers the convective heat transfer between fluid and solid, ignoring the roughness effect of 3D-printed microchannels [32]. Therefore, the simulation and experiments in this article can ensure the accuracy of the data.

### 4.3. Flow Characteristics of Heat Sinks

The geometric structure of microchannels has a significant impact on fluid flow. Figure 7 shows the flow velocity magnitude and streamline distribution of two microchannels at different flow velocities when P=19 W. Figure 7a shows the velocity distribution cloud map of the cross-section at the entrance of the microchannel. It can be seen that the velocity at the center of the channel is the highest. This is because the closer the fluid is to the wall, the greater the viscous force it experiences. The TFMCHS generates four vortices at the inlet cross-section, mainly due to the presence of ribs and grooves interfering with the radial flow of the fluid, resulting in a pressure difference and the formation of vortices [10]. 

From Figure 7b,c, it can be seen that the presence of ribs and grooves creates a throttling effect [33]. When the fluid flows through the trapezoidal rib, the sudden decrease in inlet leads to the main flow converging towards the middle, some of the streamline being interrupted, and the overall flow being S-shaped. A large stagnation zone is generated at the rear end of the rib, resulting in lower heat transfer efficiency. At the same time, due to the presence of grooves, the cross-sectional area of the channel increases, and the velocity of the fluid flowing through the grooves slows down, resulting in a stagnation zone. A vortex is generated in the stagnation zone, pulling the hot fluid inside the groove towards the central region. This allows the cooler fluid at the center and the hotter fluid near the microchannel wall to fully mix, promoting temperature uniformity [34]. The combination of ribs and grooves causes periodic interruption and generation of the boundary layer, significantly enhancing the heat transfer effect. And as the inlet velocity increases, it significantly leads to stronger flow mixing.

Figure 8 shows the variation of pressure drop with flow rate for two types of radiators when P=19 W. It can be seen that the pressure drop of the two types of radiators increases with the increase in flow rate, and the pressure drop of the TFMCHS is greater than that of the RMCHS. This is because the pressure drop in a straight channel is mainly caused by friction loss on the inner wall of the channel, and the static pressure gradually decreases along the flow direction. For complex channels, a pressure drop not only includes friction loss along the inner wall of the channel but also includes pressure loss caused by direct fluid impact on ribs and grooves [35]. As the flow rate increases, the impact effect becomes stronger, and the pressure drop loss increases.

### 4.4. Heat Transfer Characteristics of Heat Sinks

For chip cooling, the highest and average temperatures on the surface of the heat sink are of utmost concern, as the uniformity of temperature distribution directly affects the reliability, efficiency, and service life of microchips. Figure 9 and Figure 10 show the average temperature and maximum temperature of two heat sinks measured under four different power levels as a change in flow rate. Five thermocouple test points were set up on the bottom surface of the heat sink using a multichannel thermometer. The average and maximum temperatures of five data points were obtained. The test results indicate that the average and maximum temperatures of both types of radiators decrease with increasing flow rates. As the flow rate increases, the trend of decreasing average and maximum temperatures decreases, indicating that increasing the flow rate to lower the bottom temperature after reaching a certain value is not an effective method. From the figure, it can be seen that the heat dissipation capacity of TFMCHSs is significantly higher than that of RMCHSs, and as the power increases from 12 W to 33 W, the heat dissipation performance of TFMCHSs becomes better and better. When P=33 W, compared with RMCHSs, the average and maximum temperatures of TFMCHSs decreased by 6.8 °C and 9.7 °C, respectively. This is of great significance for improving the reliability of chips and extending their lifespan. 

It can be explained as follows: the combination of grooves and ribs is beneficial for enhancing heat transfer, mainly due to the interruption and redevelopment of the velocity boundary layer and thermal boundary layer, strong fluid disturbance, and the mixing effect of internal eddies. The trapezoidal rib changes the mainstream flow direction, weakening the vortex zone inside the cavity, impacting the boundary layer, and enhancing the heat transfer effect. Meanwhile, due to the presence of grooves, the convective heat transfer area inside the channel increases. Compared with straight channels, the microchannel proposed in this article has a lower and more uniform temperature and a more significant heat dissipation effect.

The temperature cloud maps of the bottom surfaces of the two radiators when $Q_{V}=30\;\mathrm{mL}/\min$ and P=19 W are shown in Figure 11. From the figure, due to the almost absence of fluid mixing in the straight passage, the temperature of the RMCHS is significantly higher than that of the TFMCHS. This is mainly because the flow inside the channel is laminar, and in the absence of external interference, there is basically only heat transfer inside the fluid, and thermal convection can be ignored. The closer the fluid is to the wall, the greater the influence of the boundary layer and the worse the heat transfer effect. Compared with the RMCHS, the TFMCHS has a much smaller temperature gradient and a more uniform temperature distribution. This indicates that ribs and grooves have a significant impact on the temperature field [36].

### 4.5. Performance Evaluation

The TFMCHS has good heat transfer characteristics and a more uniform temperature distribution, but the pressure drop also increases. Therefore, we use a thermal enhancement factor to evaluate the comprehensive performance of flow and heat transfer. The comparison of the Nusselt number $\left(Nu/Nu_{0}\right)$ and apparent friction coefficient $\left(f/f_{0}\right)$ are two parameters used to identify heat dissipation performance and pressure drop performance. From Figure 12, it can be seen that both $Nu/Nu_{0}$ and $f/f_{0}$ increase with the increase in inlet flow rate. This indicates that the TFMCHS has good heat dissipation performance, but the rib structure leads to significant friction losses [37].

The relationship between the thermal enhancement factor and flow rate when P=19 W is shown in Figure 13. From the graph, it can be seen that the enhancement coefficient is always greater than 1 and increases with the increase in inlet flow rate. This means that the comprehensive performance of the TFMCHS has been improved. When the flow rate is 32.5 mL/min, the thermal enhancement factor can reach 1.26. Therefore, for chip cooling, the TFMCHS is more economical and effective.

## 5. Conclusions

The heat transfer characteristics of TFMCHSs were studied through experiments and numerical simulations, and the fluid flow characteristics of RMCHSs were studied through numerical simulations. The conclusions are as follows: The average and maximum temperatures of the bottom surfaces of two types of heat sink at different power levels were experimentally measured. The temperature of TFMCHSs is significantly lower than that of RMCHSs. And as the power increases, the heat dissipation effect of TFMCHSs becomes stronger.Compared to RMCHSs, TFMCHSs have the highest friction loss, mainly due to the direct impact of fluid on the rib structure. The increase in flow area caused by grooves has little effect on friction loss.The combination of grooves and ribs is beneficial for enhancing heat transfer, mainly due to the interruption and redevelopment of the velocity boundary layer and thermal boundary layer, strong fluid disturbance, and the mixing effect of internal eddies. The trapezoidal rib causes the mainstream to flow in an S-shape, thereby impacting the upper and lower wall boundary layers and enhancing heat transfer. The fan-shaped groove increases the contact area between fluid and solid in the microchannel while creating a vortex zone that weakens heat conduction within the fluid but enhances convective heat transfer. This enables sufficient mixing of fluids in the channel.The comprehensive performance can be evaluated using the thermal enhancement coefficient. When the inlet flow rate is 32.5 mL/min, the thermal enhancement factor reaches 1.26. Therefore, for chip cooling, the TFMCHS is more effective and economical.

## Figures and Tables

**Figure 1 micromachines-15-00713-f001:**
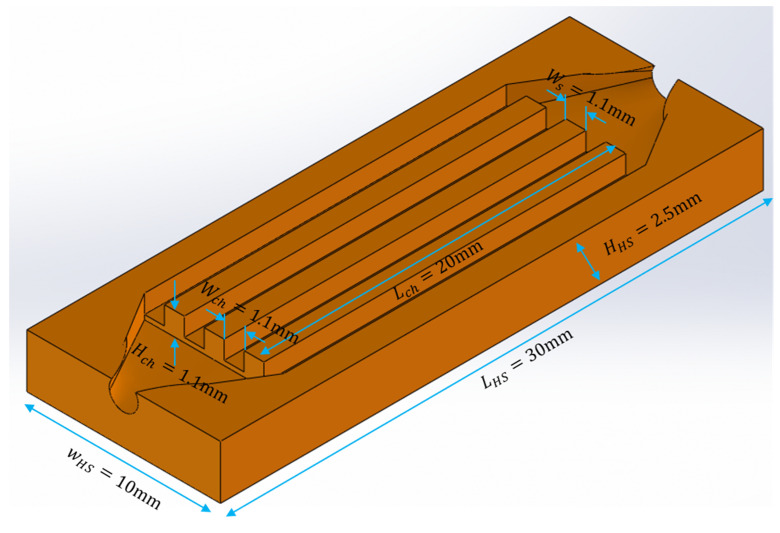
Isometric view of straight-channel heat sink test piece and its dimensions.

**Figure 2 micromachines-15-00713-f002:**
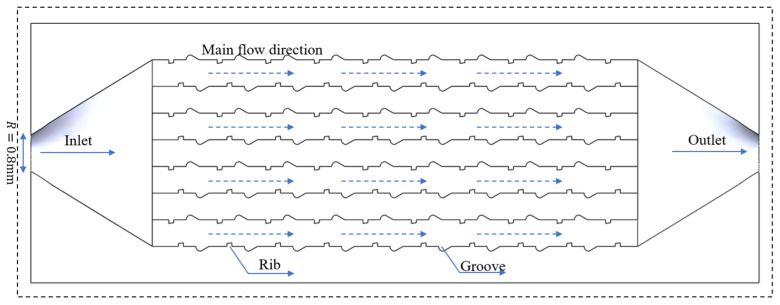
Complex-channel cross-sectional view and water inlet flow direction.

**Figure 3 micromachines-15-00713-f003:**
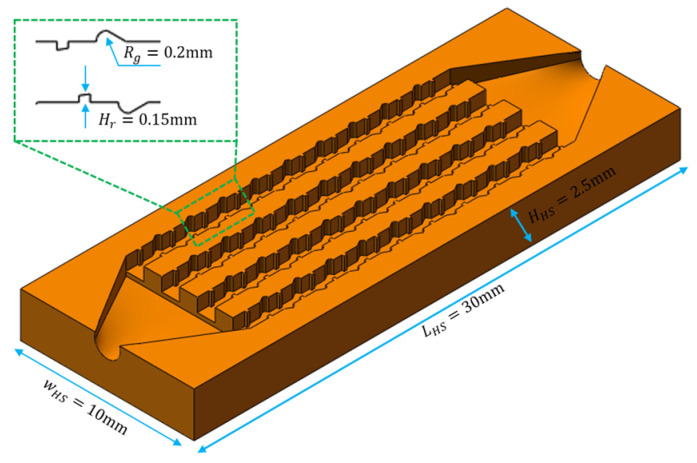
Isometric view of complex heat sink test piece and its dimensions.

**Figure 4 micromachines-15-00713-f004:**
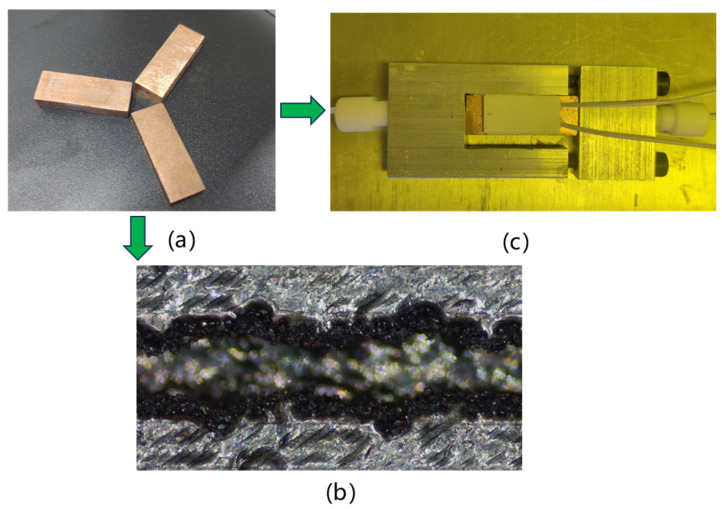
Heat sinks and packaging test. (**a**) Heat sink, (**b**) SEM images of microchannels, (**c**) packaging of heat sink.

**Figure 5 micromachines-15-00713-f005:**
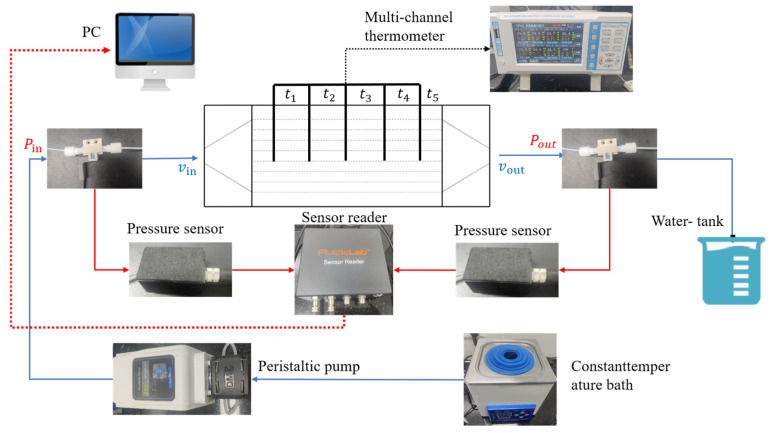
Schematic diagram of experimental apparatus.

**Figure 6 micromachines-15-00713-f006:**
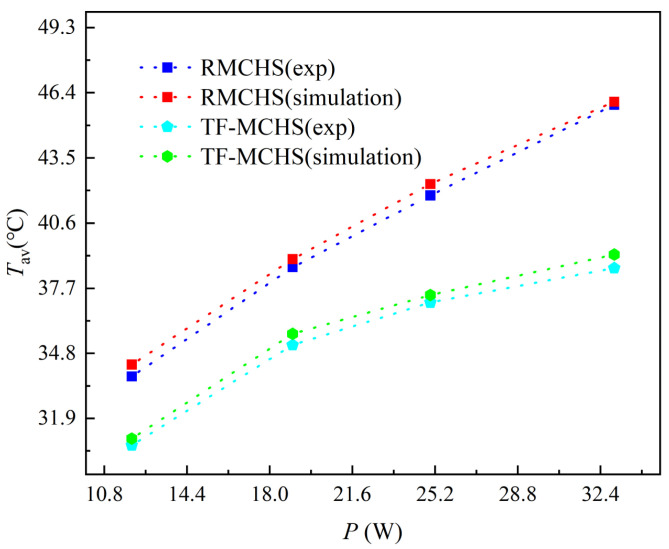
Comparison of measured and numerical solutions for the average surface temperature of heat sinks under different heating powers.

**Figure 7 micromachines-15-00713-f007:**
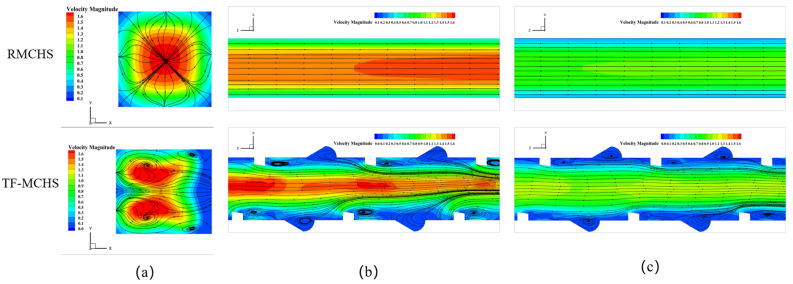
Flow velocity and streamline distribution figures for different channels: (**a**) the channel inlet for v=1 m/s; (**b**) z = 6.4–11.8 mm (y = 0.5 mm), v=1 m/s; (**c**) z = 6.4–11.8 mm (y = 0.5 mm), v=0.6 m/s.

**Figure 8 micromachines-15-00713-f008:**
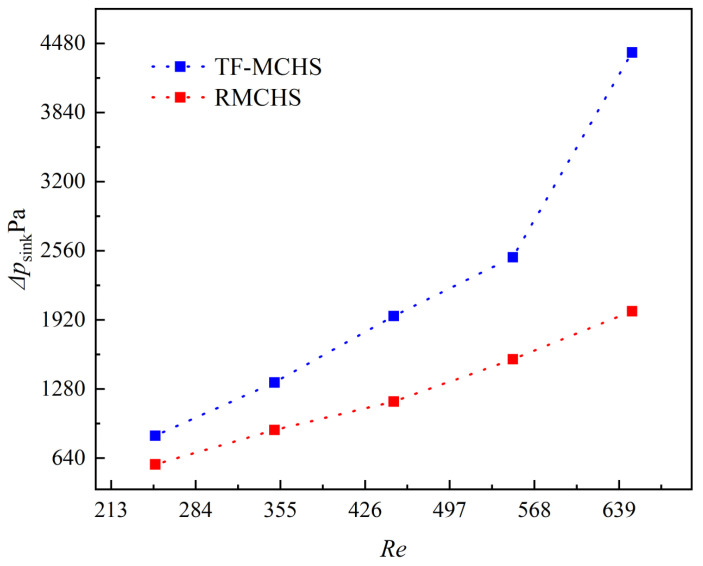
Variation in pressure drops with flow rate for two heat sinks, P=19 W.

**Figure 9 micromachines-15-00713-f009:**
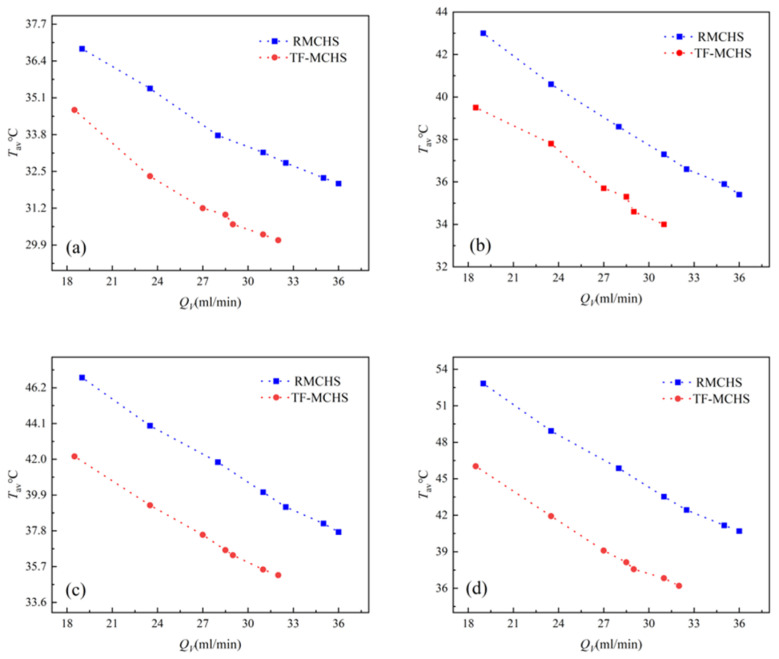
Average temperature figures of the bottom surfaces of two channels under different heating powers: (**a**) 12 W. (**b**) 19 W. (**c**) 15 W. (**d**) 33 W.

**Figure 10 micromachines-15-00713-f010:**
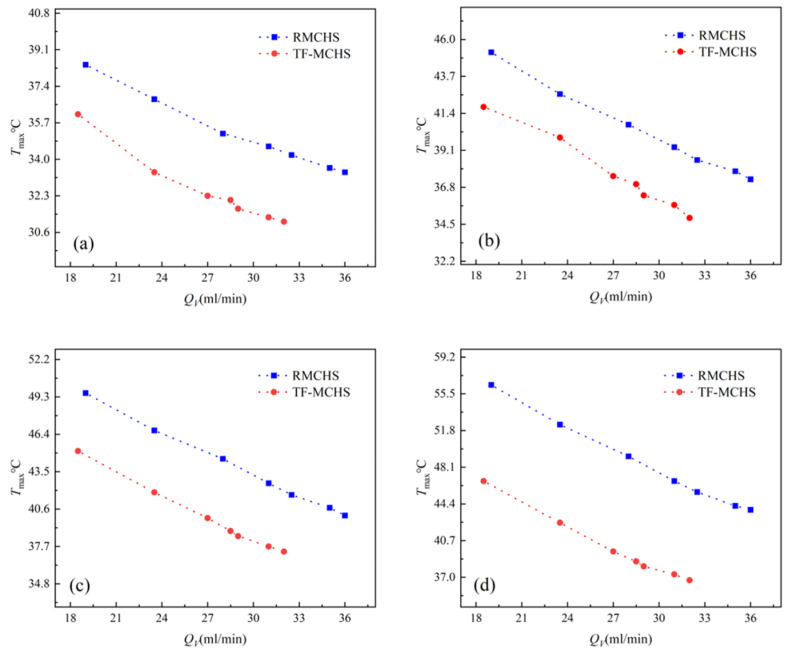
Maximum temperature figures of the bottom surfaces of two channels under different heating powers: (**a**) 12 W. (**b**) 19 W. (**c**) 15 W. (**d**) 33 W.

**Figure 11 micromachines-15-00713-f011:**
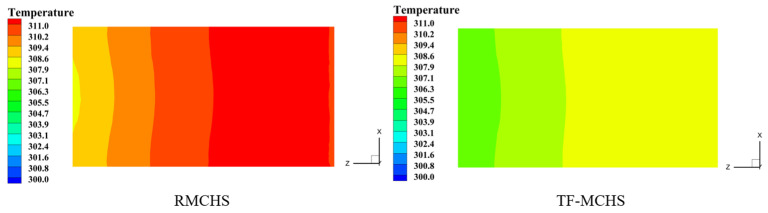
Temperature contours of the heater surface for two heat sinks, $Q_{V}=30\;\mathrm{mL}/\min$ and P=19 W.

**Figure 12 micromachines-15-00713-f012:**
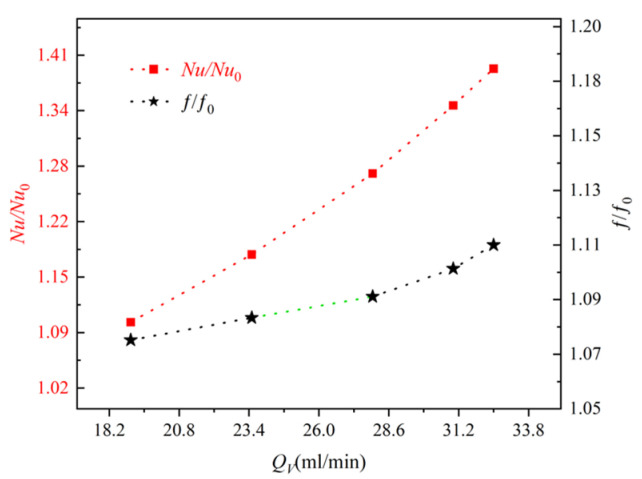
The variation of $Nu/Nu_{0}$ and $f/f_{0}$ with $Q_{V}$ in microchannel heat sinks.

**Figure 13 micromachines-15-00713-f013:**
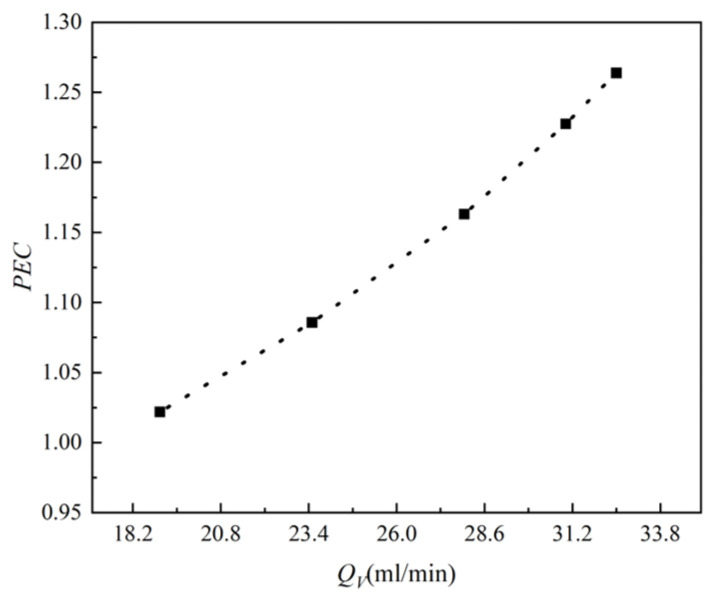
Relationship between thermal enhancement factor and flow rate for TFMCHS.

**Table 1 micromachines-15-00713-t001:** Maximum uncertainty of measured and calculated parameters.

Parameters	Maximum Uncertainty (%)	Parameters	Maximum Uncertainty (%)
$D_{h}$	1.37	Re	2.03
$Q_{v}$	2.1	Δp	1.98
P	0.327	$T_{av}$	1.45

**Table 2 micromachines-15-00713-t002:** Check of grid independence.

Case	Element Number ( )	$T_{av}$ (°C)	Err (%)
Grid 1	20.38	37.81	4.1
Grid 2	37.92	38.74	1.8
Grid 3	53.29	39.13	0.9
Grid 4	86.34	39.49	/

## Data Availability

The original contributions presented in the study are included in the article, further inquiries can be directed to the corresponding authors.

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
