# Peer review of "Experimental and Numerical Study of a Trapezoidal Rib and Fan Groove Microchannel Heat Sink"

_micromachines, 2024, doi:10.3390/mi15060713_

Round 1

Reviewer 1 Report

Comments and Suggestions for Authors

1. The applications scenario of this study should be disscussed.

2. The detailed fabrication process should be disscussed.

3. How the channel morphology was descied in the design process.

4. I am wondering if any comparsion was made between this study with other designs.

5. From the SEM image, the surface seems to be rough, authors should disscus this.

6. What is "P W" means in figure 6?

7. The flow can't be obeserved directly with substrate and cover plate are all metal, how to compare the experimental result with the simulation result?

8. Title is misleading, should clearly indicate "heat sink" in the title.

Reviewer 2 Report

Comments and Suggestions for Authors This manuscript introduces an innovative design for a microchannel heat sink (TF-MCHS) featuring trapezoidal ribs and fan-shaped grooves, which is commendable for its potential to enhance heat transfer through the periodic disruption and regeneration of boundary layers. The utilization of selective laser melting technology, a sophisticated additive manufacturing process, allows for the fabrication of intricate microchannel geometries. The integration of experimental measurements with numerical simulations to assess the TF-MCHS's performance across a range of conditions is contributes to the field of microchannel heat sink evaluation.   However, there are opportunities to further strengthen the research presented in the manuscript:   This manuscript note a good correspondence between experimental and simulation outcomes. More granular comparison, complemented by statistical analyses that quantify the discrepancies between the two sets of data, would be advantageous.   This manuscript primarily contrasts the TF-MCHS with a conventional rectangular microchannel heat sink (RMCHS). Expanding this comparison to include a wider array of contemporary heat sinks could offer a more nuanced perspective on the TF-MCHS's performance.   This manuscript does not delve into a detailed cost-effectiveness analysis of the TF-MCHS. Given that the fabrication process involves the use of CuCrZr powder and selective laser melting technology, which could entail higher costs compared to traditional manufacturing approaches.   Addressing these points would significantly enhance the depth and impact of the research, providing a more comprehensive understanding of the TF-MCHS's viability and potential for application in practical scenarios. Comments on the Quality of English Language

The English language used in the paper is clear, professional, and well-suited for the technical nature of the research, effectively conveying the study's complex findings and methodologies.

Reviewer 3 Report

Comments and Suggestions for Authors

Overall the paper presented is good. The main body of the report (methodology/results/disscussion) is well presented. Minor recommendations for the introduction section, see attached file for comments.

Round 2

Reviewer 1 Report

Comments and Suggestions for Authors

Thanks for your reply.